

# Wave energy attenuation in fields of colliding ice floes. Part B: A laboratory case study

Agnieszka Herman[1], Sukun Cheng[2], and Hayley H. Shen[3]

[1]Institute of Oceanography, University of Gdansk, Poland
[2]Nansen Environmental and Remote Sensing Center, Bergen, Norway
[3]Department of Civil and Environmental Engineering, Clarkson University, Potsdam, NY, USA

**Correspondence:** Agnieszka Herman (oceagah@ug.edu.pl)

**Abstract.** This work analyzes laboratory observations of wave energy attenuation in fragmented sea ice cover composed of interacting, colliding floes. The experiment, performed in a large (72 m long) ice tank, includes several groups of tests in which regular, unidirectional, small-amplitude waves of different periods were run through floating ice with different floe sizes. The vertical deflection of the ice was measured at several locations along the tank, and video recording was used to document the overall ice behavior, including the presence of collisions and overwash of the ice surface. The observational data are analyzed in combination with the results of two types of models: a model of wave scattering by a series of floating elastic plates, based on the matched eigenfunction expansion method (MEEM); and a coupled wave–ice model based on discrete-element model (DEM) of sea ice and a wave model solving the stationary energy transport equation with two source terms, describing dissipation due to ice–water drag and due to overwash. The observed attenuation rates are significantly larger than those predicted by the MEEM model, indicating substantial contribution from dissipative processes. Moreover, the dissipation is frequency dependent, although, as we demonstrate on the example of two alternative theoretical attenuation curves, the quantitative nature of that dependence is difficult to determine and very sensitive to assumptions underlying the analysis. Similarly, more than one combination of the parameters of the coupled DEM–wave model (restitution coefficient, drag coefficient, overwash criteria) produces spatial attenuation patterns in good agreement with observed ones over a range of wave periods and floe sizes, making selection of "optimal" model settings difficult. The results demonstrate that experiments aimed at identifying dissipative processes accompanying wave propagation in sea ice and quantifying the contribution of those processes to the overall attenuation require simultaneous measurements of many processes over possibly large spatial domains.

## 1 Introduction

This is the second part of a two-part paper in which we analyze energy attenuation of waves propagating through sea ice composed of densely packed, colliding ice floes. In the first paper (Herman *et al.* 2019, *The Cryosphere Discuss.*, paper ID tc-2019-121), referred to further as Part A, we formulated equations of a coupled, one-dimensional wave–ice model, combining a discrete-element model (DEM) of sea ice and a simple wave model based on the energy transport equation with two source terms, describing energy dissipation due to ice–water drag and due to overwash. We analyzed theoretically the solutions of the model equations in the limiting case of a compact, horizontally constrained ice cover, demonstrating that the model predicts





non-exponential wave attenuation, with attenuation rates strongly dependent on the wave group velocity, i.e., on dispersion relation – and thus on ice type. We also performed a detailed analysis of the model sensitivity to several parameters, including ice–water drag coefficient, restitution coefficient, and floe size. In general, the simulated wave amplitude profiles reflect the existence of two zones with very different dynamics – a narrow zone at the ice edge with energetic collisions and very strong

attenuation, and an inner zone with densely packed ice floes undergoing limited horizontal displacements, with attenuation rates close to those derived theoretically for compact ice.

The second part of the study, described in this paper, is based on observational data from a laboratory experiment in a large ice tank, in which regular, unidirectional waves were run through ice covers composed of rectangular floes of equal size. The experiments cover a range of floe lengths and wave periods, and include measurements of the vertical deflection of the ice

by means of underwater pressure sensors and motion tracking methods, as well as video recordings of the ice motion. The laboratory measurements, combined with results of two numerical models – a model of nondissipative scattering by a series of floating, elastic plates, and a coupled DEM–wave model described in Part A – are analyzed in order to get insights into processes contributing to attenuation of wave energy in fields of colliding, interacting ice floes.

Floe–floe collisions are often mentioned in the literature as one of several mechanisms contributing to wave energy dissi-

pation in the marginal ice zone (MIZ). However, field observations of colliding ice floes are rare, and measurements directly relating collisions to dynamical processes in sea ice and underlying surface layer of the ocean are practically nonexistent. The first studies devoted to ice floe collisions, conducted in the 1980s and early 1990s, were based on measurements with accelerometers placed on the ice (Martin and Becker, 1987, 1988; Martin and Drucker, 1991; McKenna and Crocker, 1992; Rottier, 1992) or with the help of so-called "strain arrays" (e.g., Hibler III and Leppäranta, 1984). The second method was

particularly suitable for detection of low-frequency collisions related to larger-scale shear deformation of the ice cover (e.g., Shen et al., 1984) and provided observations that formed the basis for formulating collisional rheology models for the MIZ (Shen et al., 1986, 1987; Lu et al., 1989). The accelerometer-based observations, more relevant from the point of view of this study, concentrated mainly on high-frequency collisions related to the forcing of the ice by waves and inspired numerical studies with DEM models by Shen and Ackley (1991), Frankenstein and Shen (1993), Hopkins and Shen (2001) and Shen and

Squire (1998). As discussed in Part A, the work by Shen and Squire (1998) is particularly important for the present study, as it concentrates on mechanisms dissipating the energy of waves propagating through broken sea ice. Contrary to those earlier modelling studies, relevant for very small ice floes (e.g., pancakes floating on long-period swell), the recent DEM model by Herman (2018) concentrated on wave-induced surge motion and collision patterns of large floes, with sizes comparable with wavelength. Finally, collisions between two ice floes moving on waves have been studied in the laboratory by Yiew et al.

(2017). Li and Lubbad (2018) analyzed kinematics of colliding ice floes based on data from the same experiment that is used in this paper. Hitherto laboratory experiments devoted to wave attenuation in ice concentrated mostly on frazil, grease and pancake ice or mixtures of those ice types, that is, conditions in which collisions were insignificant (see, e.g., Zhao and Shen, 2015; Rabault et al., 2019; Yiew et al., 2019, and references there). The same is true for most MIZ field studies (e.g., Rogers et al., 2016; De Santi et al., 2018; Voermans et al., 2019). Although fragmented ice was among ice types considered by Zhao





and Shen (2015) and Yiew et al. (2019), their ice covers consisted of a layer of relatively small, overlapping floes surrounded by a dense ice–water mixture, in which collisions didn't play any noticeable role.

As already mentioned, the ice cover analyzed in this study consists of rectangular, densely packed ice floes. As the video documentation shows, the floes undergo regular collisions and, in tests with relatively high wave steepness, overwash, espe-
cially in the zone close to the ice edge, indicating dissipation mechanisms that are presumably relevant for wave attenuation. However, apart from incoming wave characteristics and the basic ice properties, the only quantitative information available is the wave amplitude at several locations along the tank. The main goal of this study is to demonstrate that the interpretation of the observed attenuation and validation of numerical models based on that type of data is problematic, as many mutually interrelated mechanisms contribute to the net attenuation. This is important, because a situation in which wave amplitude
data are available without additional information on dissipation is a rule rather than an exception. In particular, satellite data are increasingly used to assess wave attenuation in sea ice, without complementary information on processes taking place in and under the ice. Moreover, many different models can be calibrated to reproduce observations with reasonable accuracy, especially considering large uncertainties in attenuation rates derived from measurements.

In the next section, we provide a brief description of the two models used (more details concerning the coupled DEM–wave
model can be found in Part A), followed by a description of the laboratory experiment in section 3. We then move, in section 4, to the analysis of observed wave attenuation in combination with the results of the MEEM model. We show that in the large majority of tests the scattering model does not explain the observed attenuation. This result indicated that dissipative processes have a large contribution to the overall attenuation. In order to quantitatively describe the attenuation rates at different wave frequencies and floe sizes, the observational data are fitted with two alternative theoretical attenuation curves, the exponential
function considered in most similar studies, and the function resulting from dissipation due to ice–water drag, discussed in Part A. We show that the available data are not sufficient to select any of the two functions as "better" than the other, and that the resulting attenuation coefficients are extremely sensitive to the initial assumptions regarding the incident wave amplitude, as well as to location of the individual data points. We then move to the analysis of DEM simulations and show that more than one combination of model parameters (including the ice–water drag coefficient, restitution coefficient, and parameters describing
the occurrence and intensity of overwash) produces attenuation patterns similar to the observed ones. We summarize the results and discuss their consequences in section 6.

## 2   Numerical models

As mentioned in the introduction, two very different numerical models are used in this work to aid our understanding of processes observed in the laboratory. The first model (section 2.1) simulates scattering, but disregards all dissipative processes.
The second model (section 2.2) disregards scattering, but takes into account dissipation resulting from combined effects of floe collisions and ice–water drag, as well as from overwash.





## 2.1 Model of wave attenuation due to scattering

In order to analyze the non-dissipative attenuation processes in the setup considered, we use the matched eigenfunction expansion method (MEEM) by Kohout et al. (2007); Kohout (2008). In this model, ice is represented as a series of elastic plates floating on the sea surface and staying in contact with each other (i.e., there are no open water spaces between neighboring floes). The plates do not move horizontally, but they undergo vertical deflection due to water motion underneath. The waves are assumed time-harmonic, linear and irrotational, so that they can be described by a velocity potential $\Phi(x,t)$, the spatial component of which, $\phi(x)$, is represented, for each plate, as a sum of transmitted and reflected propagating, damped propagating, and evanescent modes. The amplitude of each mode is determined from the kinematic and dynamic boundary conditions at the bottom, ice-free or ice-covered regions, and vertical edges of the floes, complemented with a requirement that $\phi$ and $d\phi/dx$ are continuous at plate boundaries. Importantly, the boundary conditions in the ice-covered region are formulated based on the elastic plate dispersion relation. The assumptions of the MEEM model make it suitable for confined ice (insignificant horizontal ice motion, no collisions between ice floes) and for relatively large floes, with sizes comparable with wavelength. Due to those assumptions, the model tends to overestimate attenuation of high-frequency waves and to underestimate attenuation of low-frequency waves (Kohout and Meylan, 2008; Kohout et al., 2011). All equations can be found in Kohout et al. (2007) and Kohout (2008).

In the analysis in this paper, we make use of two quantities computed with the MEEM model: the amplitude of the vertical deflection of the ice corresponding to the transmitted propagating mode ($T_0$ in the notation of Kohout et al., 2007), which we denote with $a_{\mathrm{MEEM},T_0}$; and the total amplitude, including the contributions of the transmitted and reflected propagating, damped propagating, and the first 30 evanescent modes (i.e., $T_{-2}, T_{-1}, T_0, T_1, \ldots, T_{30}, R_{-2}, R_{-1}, R_0, R_1, \ldots, R_{30}$), which we denote with $a_{\mathrm{MEEM,tot}}$. Whereas $a_{\mathrm{MEEM},T_0}$ is constant over a given floe, $a_{\mathrm{MEEM,tot}}$ may strongly vary within a floe (see further section 4).

## 2.2 A discrete-element sea ice model with wave dissipation

The model used in this study is described in detail in Part A. Here we only recap its main features and summarize its behavior.

The model consists of two coupled parts: a DEM sea ice model (based on Herman, 2016, 2018), simulating the motion and interactions of individual ice floes, and a wave energy transport model, simulating wave propagation and attenuation in sea ice. The coupled model is one-dimensional, i.e., it computes propagation of uni-directional waves through a series of floes arranged along the $x$ axis, numbered $i = 1, 2, \ldots, N_f$, and the only component of the ice motion considered is the surge and drift along that axis, i.e., the relevant time-dependent variables for each floe are the horizontal position of its center of mass $x_i$ and its horizontal translational velocity $u_i$. The ice floes are cuboid and have identical thickness $h_i$, length in the wave propagation direction $L_x$, density $\rho_i$ and material properties: elastic modulus $E$, Poisson's ratio $\nu$ and restitution coefficient $\varepsilon$. The DEM model solves the linear-momentum equations for each ice floe, with four types of forces: the wave-induced Froude–Krylov force $F_{\mathrm{w},i}$, the virtual (or added) mass force $F_{\mathrm{v},i}$, the drag force $F_{\mathrm{d},i}$, and the sum of contact forces from all collision/contact partners of floe $i$, $F_{\mathrm{c},i}$.



Propagating, regular waves are assumed with known period $T = 2\pi/\omega$ and length $L_w = 2\pi/k$ (where $\omega$ denotes the angular frequency and $k$ the wavenumber), with a dispersion relation $\omega(k)$ and group velocity $c_g = \mathrm{d}\omega/\mathrm{d}k$. The wave amplitude $a = a(x)$ is computed from the stationary wave energy conservation equation, assuming a known incident amplitude $a_0$ at the ice edge ($x = x_0$), with two (negative) source terms: dissipation due to ice–water skin drag $S_{\mathrm{sd}}$ and due to overwash $S_{\mathrm{ow}}$. The

first dissipation term is related to non-zero relative ice–water velocity; a quadratic drag law is assumed with a constant drag coefficient $C_{\mathrm{sd}}$. For the overwash dissipation, a very unsophisticated parametrization is used, in which the overwash is treated as a shallow water wave with average depth $h_{\mathrm{ow}}$ (Skene et al., 2018), and $h_{\mathrm{ow}}$ is assumed proportional to the wave steepness $(ka)$, with $h_{\mathrm{ow}} > 0$ if a certain minimal steepness $s_{\mathrm{min}}$ is exceeded:

$$h_{\mathrm{ow}} = c_{\mathrm{ow}} \max\{ka - s_{\mathrm{min}}, 0\}, \tag{1}$$

with $c_{\mathrm{ow}}$ an adjustable parameter.

The coupled model is solved with an iterative algorithm, in which the sea ice and wave modules are run in turns until a stationary wave amplitude profile $a(x)$ is reached.

As analyzed in detail in Part A, for compact, horizontally confined sea ice the model predicts attenuation of the form:

$$a(x) = \frac{1}{\alpha_c x + 1/a_0}, \quad \text{with} \quad \alpha_c = \frac{4C_{\mathrm{sd}}}{3\pi g} \frac{\omega^3}{c_g \tanh^3[kh]}, \tag{2}$$

where $h$ denotes water depth and $g$ acceleration due to gravity. Thus, attenuation is non-exponential and, for small $x$, dependent on the incident wave amplitude $a_0$. When collisions are present, the shape of the simulated attenuation curves $a(x)$ reflects the existence of two regions, a narrow zone of energetic collisions and strong attenuation close to the ice edge, and an inner zone of densely packed floes with attenuation rates close to those described by (2).

## 3 Laboratory observations of wave attenuation in fragmented ice

The experiments analyzed in this work were performed in the Large Ice Model Basin (LIMB) of the Hamburg Ship Model Basin (Hamburgische Schiffbau-Versuchsanstalt, HSVA) as part of the Hydralab+ Transnational Access project "Loads on Structure and Waves in Ice" (LS-WICE; project under the Horizon 2020 EU Framework Programme for Research and Innovation, H2020-INFAIA-2014-2015). Initial results of the tests relevant for this study (series 2000 and 3000, as described below) are described in Cheng et al. (2017). Recently, Cheng et al. (2018) used the same data in an analysis of the influence of floe size on

wave dispersion. Other LS-WICE results were used to study floe-size distributions in sea ice broken by waves (Herman et al., 2017, 2018), as well as wave-induced collisions and floe kinematics (Li and Lubbad, 2018).

A sketch of the experiment setup is shown in Fig. 1; Table 1 provides a summary of tests analyzed in this study: the wavemaker wave amplitude $a_{0,w}$ and wave period $T$, as well as floe length $L_x$. In both test groups, the ice sheet was cut into six "stripes" (in the direction parallel to the tank axis) with equal width, further referred to as floe rows and indexed $j = 1, \ldots, 6$.

The floes within each row are numbered $i = 1, \ldots, N_f$, with $N_f$ depending on the floe length $L_x$ (see further). Details related to the preparation of the ice sheets and conduction of the tests can be found in Cheng et al. (2018) and won't be repeated here. In this section, we only provide information that is relevant for the present study.





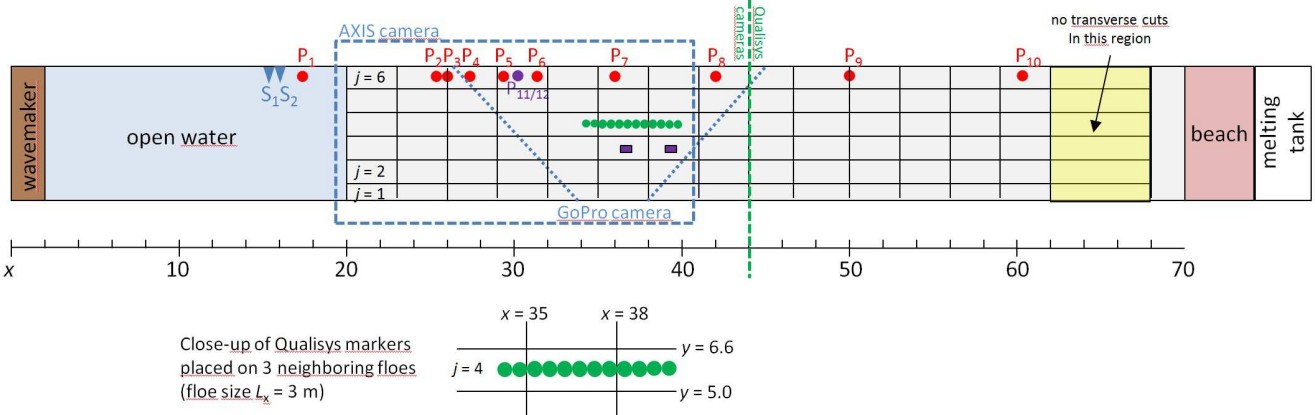

**Figure 1.** LS-WICE experiment setup. The ice edge is located at $x_0 = 20$ m, the gray area represents the ice sheet (at the stage when it was cut into floes with $L_x = 3$ m). Red dots numbered $P_1, \ldots, P_{10}$ show positions of pressure sensors, point $P_{11}/P_{12}$ – position of a double pressure sensor (measuring at two different depths), $S_1$ and $S_2$ – positions of ultrasound sensors. The dashed and dotted blue lines mark the fields of view of the AXIS camera mounted at the ceiling and the GoPro camera mounted at the side of the basin. Green dots mark the Qualisys markers placed on the 4th row of ice floes (shown in more detail in the schematic below the main figure), and violet rectangles show the two IMUs placed on the 3rd row of floes.

The two ice sheets, used in series 2000 and 3000, respectively, had the same ice thickness $h_i = 0.036$ m, but slightly different ice density and, especially, different elastic moduli (Table 1). No measurements of the restitution coefficient $\varepsilon$ of the ice were done during LS-WICE. Although Li and Lubbad (2018) attempted to estimate $\varepsilon$ from the pre- and post-collisional floe velocities based on data from test 3210, their results are likely strongly underestimated due to the fact that the surface convergence related to the wave motion prohibited the floes from separating from each other after collisions. In this study, therefore, we treat the restitution coefficients as unknown.

Crucially for floe motion and collisions, a floating boom was installed at the ice edge ($x = 20$ m) in both tests, preventing the ice floes from drifting in the up-wave direction. The ends of the boom were fixed to the side walls of the tank, but in some tests the central section of the boom bent under the pressure of the ice, so that the average floe–floe distance along the tank axis (rows 3 and 4) was slightly larger than along the walls (rows 1 and 6). Visual observations showed that the horizontal ice motion at the opposite, down-wave end of the ice sheet, close to the parabolic beach installed there, was very limited.

Each test series consisted of three groups of tests, with floe lengths $L_x$ equal to 6 m, 1.5 m and 0.5 m in series 2000, and 3 m, 1.5 m and 0.5 m in series 3000 (Table 1). Each test group (with an exception of tests 2410–2460) included runs with at least six different wave periods. In all cases the waves can be regarded as low-amplitude, deep-water waves: the largest wave steepness $ka_0$ equaled 0.07 in test 2760, and the smallest value of $\tanh[kh]$ occurred in test 3370 and equaled 0.96. The ratio of wavelength $L_w$ (computed based on wavenumbers determined from measurements) to floe length $L_x$ varied from below 1



**Table 1.** Summary of the setup of experiments in test series 2000 and 3000 (in chronological order). The ratio $L_w/L_x$ was computed for wavenumbers determined from observations (Fig. 2). See text for comments on entries in italics, marked with a star.

| **Series 2000** ($E = 2.97 \cdot 10^7$ Pa, $\rho_i = 919$ kg/m$^3$) | | | | | | **Series 3000** ($E = 5.64 \cdot 10^7$ Pa, $\rho_i = 916$ kg/m$^3$) | | | | | |
|---|---|---|---|---|---|---|---|---|---|---|---|
| Test ID | $L_x$ (m) | $T$ (s) | $a_{0,w}$ ($10^{-3}$ m) | $L_w/L_x$ (–) | comments | Test ID | $L_x$ (m) | $T$ (s) | $a_{0,w}$ ($10^{-3}$ m) | $L_w/L_x$ (–) | comments |
| 2410 | 6.0 | 2.0 | 25.0 | 1.01 | | *3110 ** | 3.0 | 2.0 | 12.5 | 2.10 | strong freezing |
| 2420 | 6.0 | 1.8 | 25.0 | 0.83 | | *3120 ** | 3.0 | 1.8 | 12.5 | 1.71 | strong freezing |
| 2460 | 6.0 | 0.9 | 5.0 | 0.24 | | *3130 ** | 3.0 | 1.6 | 12.5 | 1.40 | strong freezing |
| 2450 | 6.0 | 0.9 | 10.0 | 0.28 | | 3140 | 3.0 | 1.5 | 15.0 | 1.22 | freezing |
| 2440 | 6.0 | 2.0 | 10.0 | 0.99 | | 3150 | *3.0 ** | 1.4 | 15.0 | 1.20 | ice breaking |
| *2430* | *6.0* | *2.0* | *20.0* | – | erroneous meas. | 3160 | *3.0 ** | 1.1 | 12.5 | 0.79 | ice breaking |
| 2610 | 1.5 | 2.0 | 12.5 | 4.04 | | 3210 | 1.5 | 2.0 | 12.5 | 4.12 | |
| 2620 | 1.5 | 1.8 | 12.5 | 3.30 | | 3220 | 1.5 | 1.8 | 12.5 | 3.38 | |
| 2630 | 1.5 | 1.6 | 12.5 | 2.66 | | 3230 | 1.5 | 1.6 | 12.5 | 2.68 | |
| 2640 | 1.5 | 1.4 | 12.5 | 2.04 | | 3240 | 1.5 | 1.5 | 15.0 | 2.35 | |
| 2650 | 1.5 | 1.2 | 12.5 | 1.52 | | 3250 | 1.5 | 1.4 | 15.0 | 2.10 | |
| 2660 | 1.5 | 0.9 | 12.5 | 0.90 | | 3260 | 1.5 | 1.1 | 12.5 | 1.70 | |
| 2710 | 0.5 | 2.0 | 12.5 | 12.52 | | 3310 | 0.5 | 2.0 | 12.5 | 12.56 | |
| 2720 | 0.5 | 1.8 | 12.5 | 9.90 | | 3320 | 0.5 | 1.8 | 12.5 | 10.17 | |
| 2730 | 0.5 | 1.6 | 12.5 | 7.97 | | 3330 | 0.5 | 1.6 | 12.5 | 8.00 | |
| 2740 | 0.5 | 1.4 | 12.5 | 5.84 | | 3340 | 0.5 | 1.5 | 15.0 | 6.99 | |
| 2750 | 0.5 | 1.2 | 12.5 | 4.10 | | 3350 | 0.5 | 1.4 | 15.0 | 6.09 | |
| 2760 | 0.5 | 0.9 | 12.5 | 2.52 | | 3360 | 0.5 | 1.1 | 15.0 | 3.63 | |
| | | | | | | 3370 | 0.5 | 2.3 | 12.5 | 16.18 | |
| | | | | | | 3380 | 0.5 | 2.2 | 12.5 | 15.04 | |
| | | | | | | 3390 | 0.5 | 2.1 | 12.5 | 13.85 | |

in tests with large floes and small wave periods (e.g., 2460, 3160) to more than 10 in tests with the smallest (0.5 m) floes and long wave periods (e.g., 2710, several tests in group 3300).

The air temperature during the tests was close to 0°C; nevertheless, strong freezing was observed at the beginning of test series 3000, both between neighboring ice floes and between the ice sheet and the side walls of the tank. As a result, the ice in tests 3110–3130 (and, to a lesser extent, in test 3140) behaved as a continuous sheet rather than separate floes, and the vertical ice deflection in the central part of the tank (rows 3 and 4) exceeded the incident wave amplitude, whereas it was close to zero along the walls (rows 1 and 6), making the results of those tests of little use from the point of view of this study. Moreover, strong breaking of ice floes occurred during the course of test 3150 (see Supplementary Movie 1): in the central rows of floes ($j = 3$ and $j = 4$), the first 4 floes starting from the ice edge ($i = 1, \ldots, 4$) broke within the first 15–20 s of the test; after





roughly 1 minute, the first two floes, initially 3 m long, were broken into four pieces each. Thus, the test 3150 and especially the subsequent test 3160 in fact represent cases with floes smaller than 3 m in the area within 12 m from the ice edge. Remarkably also, as the Supplementary Movie 1 clearly shows, slightly different timing of breaking in the two neighboring rows of floes in the middle of the tank resulted in visibly different intensity of overwash, with stronger overwash over smaller floes. This

shows clearly how sensitive the behavior of the wave–ice system is to seemingly slight changes of conditions. Overall, strong overwash was observed in several tests, especially those with short waves and thus relatively high wave steepness (although, due to stronger attenuation, the region with overwash was in those cases limited to a relatively narrow zone close to the ice edge). Further down the ice sheet, starting from 15–20 m from the edge, only weak overwash was present that didn't cover the whole upper surface of the floes – to the contrary, it was limited to floes' boundaries and clearly related to closing of spaces

between floes during collisions (Supplementary Movie 2).

The wave amplitude data of interest in this study was obtained by two methods (Fig. 1): underwater pressure sensors mounted along the side wall of the tank, and, in series 3000, by the Qualisys system, recording three-dimensional position of markers placed on the ice. (In a few tests in series 3000, two inertial measurement units, or IMUs, were additionally used, placed on a row of floes neighboring the Qualisys markers, as seen in Fig. 1 and Supplementary Movies 1 and 2; however, as they

essentially duplicate the information provided by Qualisys, we don't use them in our analysis.)

The pressure sensors were located at depth of 0.35 m and distance 0.65 m from the walls. The 12 Qualisys markers were placed along the central axis of the tank (floe row $j = 4$), between $x = 34.25$ m and $x = 39.75$ m, with 0.5 m distance from each other, in such a way that in the last group of tests (3310–3390) one marker was located in the middle of each 0.5-m-long floe (Fig. 1). Details of data processing can be found in Cheng et al. (2018) and won't be repeated here.

Figure 2 shows wavenumbers $k$ determined from the pressure sensor data in both test series (see also Cheng et al., 2018). As can be seen, all values of $k$ from the tests with the smallest floes, $L_x = 0.5$ m (triangle symbols in Fig. 2), lie within the region bounded by the curves corresponding to the mass loading and open water dispersion relations. In tests with larger floes, as can be expected, the wavenumbers are lower, with values between those computed from the elastic plate and open water models. The decrease of $k$ with increasing $L_x$ occurs in all cases except in test 3160, in which, as noted above, the actual floe size was

smaller than 3 m due to ice breaking.

## 4   Observed wave attenuation

In this section we analyze the wave attenuation observed in the LS-WICE experiments, based on the data from pressure sensors and from the Qualisys system – two data sources that provide a slightly different picture of the situation due to the fact that the pressure sensors have fixed positions relative to the tank, and the Qualisys markers have fixed positions relative to the

ice. In order to better understand the observed attenuation patterns, we supplement the data analysis with vertical deflections computed from the MEEM model described in section 2.1.

The amplitudes determined from measurements and computed with MEEM are shown in Supplementary Figs. 1 and 2 and, for two selected tests, in Fig. 3. The first important observation from the amplitude profiles along the tank computed with


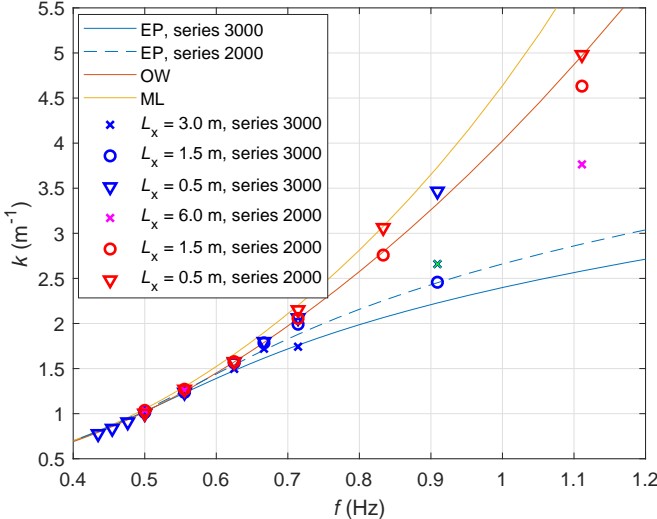

**Figure 2.** Wavenumber $k$ (m$^{-1}$) in the experiments from LS-WICE series 2000 and 3000, together with corresponding wavenumbers from the elastic plate (EP), open water (OW) and mass loading (ML) dispersion relations (in the case of ML, the curves for both series are indistinguishable). The cross symbol shown in green corresponds to test 3160, nominally with $L_x = 3.0$ m, but with smaller actual floe size due to breaking.

MEEM is that, in the majority of tests, both the total amplitude $a_{\mathrm{MEEM,tot}}$ and the amplitude of the transmitted propagating component $a_{\mathrm{MEEM},T_0}$ remain fairly constant throughout the length of the tank, suggesting that the wave attenuation due to scattering is very limited (importantly, the results do not change significantly if small, few millimeter wide gaps between ice floes are included in the MEEM setup, analogous to those present in the laboratory). The exception are tests 2640, 3110 and
5 3250, all three with floe length $L_x$ equal almost exactly half the wavelength (see $L_w/L_x$ ratios in Table 1), and, to a lesser extent, tests with the highest wave frequency, i.e., those with the lowest $L_w/L_x$ ratios. Also, for a given floe size, the ratio $a_{\mathrm{MEEM},T_0}/a_{\mathrm{MEEM,tot}}$ decreases with decreasing wave period – the contribution of the damped travelling components, leading to stronger deflection of the edges of the floes, is more significant for short waves than for long waves. Those observations are summarized (for test group 3000) in Fig. 4. In spite of the fact that the amplitude $a_{\mathrm{MEEM},T_0}$ decreases with increasing wave
frequency (Fig. 4a), i.e., at high wave frequencies a substantial part of wave energy is contained within the damped components, in the majority of tests this energy redistribution does not lead to significant wave attenuation – the transmitted amplitude at the end of the tank is very close to the initial amplitude $a_{0,w}$ in all tests except those mentioned earlier (Fig. 4c). Importantly also, although the redistribution of wave energy among the propagating and damped propagating components clearly depends on floe size, the amount of wave energy reaching the down-wave end of the tank is almost the same for all three floe sizes
considered (again, with the exception of those tests in which reflection is substantial, as seen if Fig. 4b).

Within the ice-covered region, the redistribution of wave energy among different components means that large amplitude differences can be expected within a single floe, resulting not from the overall wave attenuation, but from the contribution of





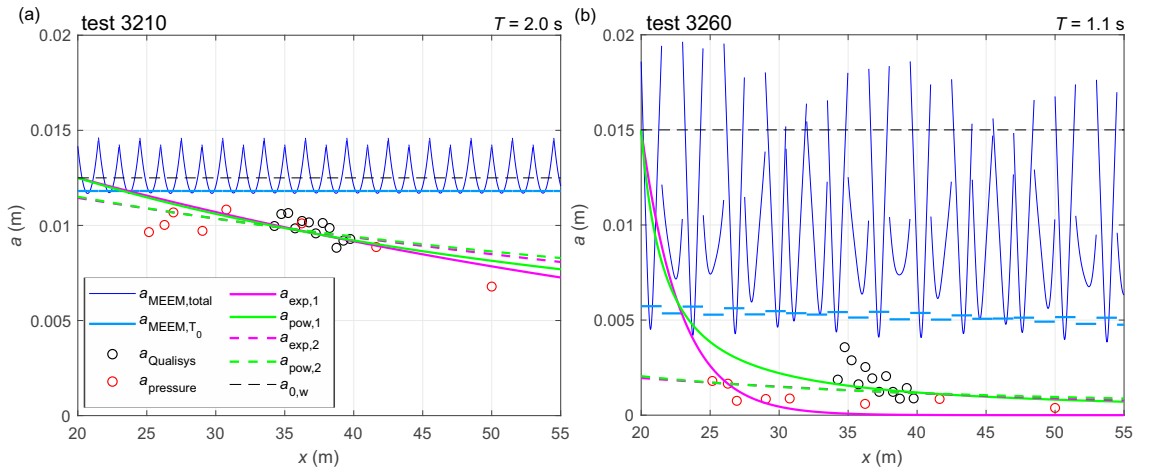

**Figure 3.** Amplitude of the vertical ice deflection along the tank in two selected tests from series 3000: 3210 (a) and 3260 (b; for analogous plots for all tests from both series, 2000 and 3000, see Supplementary Figs. 1 and 2). The measured amplitude is shown with red (pressure sensors) and black (Qualisys) circles. Thin blue lines show the total amplitude from the MEEM model; the corresponding amplitudes of the transmitted propagating mode ($T_0$) are shown with thick light blue lines. The black dashed line shows the wavemaker amplitude $a_{0,w}$. Green and magenta lines are least-square fits of the data with functions (3) and (4), respectively (continuous lines: prescribed $a_0$, dashed lines: fitted $a_0$). See text for details.

multiple wave modes at each floe edge – and this effect becomes stronger with increasing floe size. Supplementary Fig. 3, showing zoomed fragments of the tank around the location of the Qualisys markers, provides a good illustration of that variability, present both in MEEM results and, to a lesser extent, in the measured data. In consequence, the vertical deflection of the ice measured at a single point is of little value, especially if the measurement is done at a fixed position in space (as is the case

with the pressure sensors in our experiment), so that the location of the sensor relative to the floes' boundaries is unknown and might change slightly during a single test. In the case of Qualisys measurements, an attempt could be made to "split" the measured amplitudes among the transmitted propagating and the remaining modes, based on the corresponding amplitudes from the MEEM model ($a_{\mathrm{MEEM,tot}}$ and $a_{\mathrm{MEEM},T_0}$), but this approach requires several rather arbitrary assumptions, including that about the same ratio $a_{\mathrm{MEEM},T_0}/a_{\mathrm{MEEM,tot}}$ in non-dissipative waves considered by MEEM and in dissipative waves observed

in the tank. Without those assumptions, a way to proceed is to come to terms with the high scatter of the measured amplitudes (resulting from scattering processes as well as uncertainties related to measurement method and data processing), even though this means that the data provide useful information on attenuation only if, first, a large number of sensors are available, and second, if they are distributed over a long distance.

    A cursory inspection of Fig. 3 and Supplementary Figs. 1–3 is enough to conclude that the MEEM model – that is, scattering

alone – does not explain the variability of wave amplitudes observed in LS-WICE. Qualitatively, the measured values are in most tests well below those computed with MEEM, suggesting an important role of dissipative processes. Also, as can be expected, attenuation increases with increasing wave frequency – a very rough quantitative measure supporting this observation





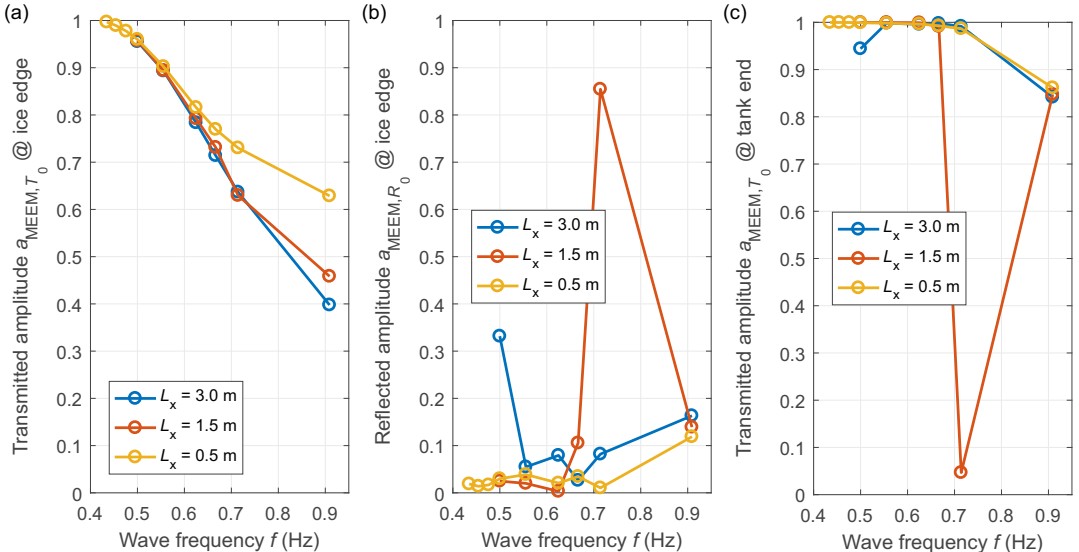

**Figure 4.** Transmission and reflection coefficients computed with the MEEM model for tests from group 3000 (the results for group 2000 are similar): amplitude of the transmitted (a) and reflected (b) propagating component at the ice edge, and of the transmitted propagating component at the end of the tank (i.e., after propagating through the whole ice sheet; c). All amplitudes are normalized with the respective $a_{0,w}$.

can be obtained by simply computing an average measured amplitude for each test (not shown). In order to get more insight into the observed attenuation, we fit the data with two theoretical curves, the exponential function:

$$a = a_0 \exp(-\alpha_{\mathrm{exp}} x_r), \tag{3}$$

and the function (2) predicted by the model of dissipation due to ice–water drag, derived in Part A:

$$a = a_0/(a_0 \alpha_{\mathrm{pow}} x_r + 1). \tag{4}$$

In (3) and (4), $x_r = x - x_0$ denotes the distance from the ice edge, located at $x = x_0$. Both fits are computed in two versions: the wave amplitude at the ice edge, $a_0$, is treated either as a freely adjustable parameter (together with $\alpha_{\mathrm{exp}}$ or $\alpha_{\mathrm{pow}}$, respectively) or is fixed to the value of the wavemaker amplitude $a_{0,w}$ used in a given test. In other words, in the first approach $a_0$ is assumed unknown and is part of the solution; the second approach might be viewed as analogous to a field case in which open-water,

incident wave height is known, e.g., from a spectral wave model or satellite observations (as, e.g., in Stopa et al., 2018), and attenuation within the ice is expressed relative to those open-water conditions (which amounts to implicitly assuming negligible reflection at the ice edge). Notably, the exponential fit with adjustable $a_0$ was applied to LS-WICE data by Cheng et al. (2018), who concluded based on the results that there was no dependence of attenuation rate on wave frequency. Indeed, treating $a_0$ as a free parameter produces fits with $\alpha_{\mathrm{exp}}$ and $\alpha_{\mathrm{pow}}$ very strongly (and apparently randomly) varying from test to test (Fig. 5a,b)

and with $a_0$ values that simply reflect the above-mentioned decrease of the area-averaged amplitude with frequency (Fig. 5c,d;



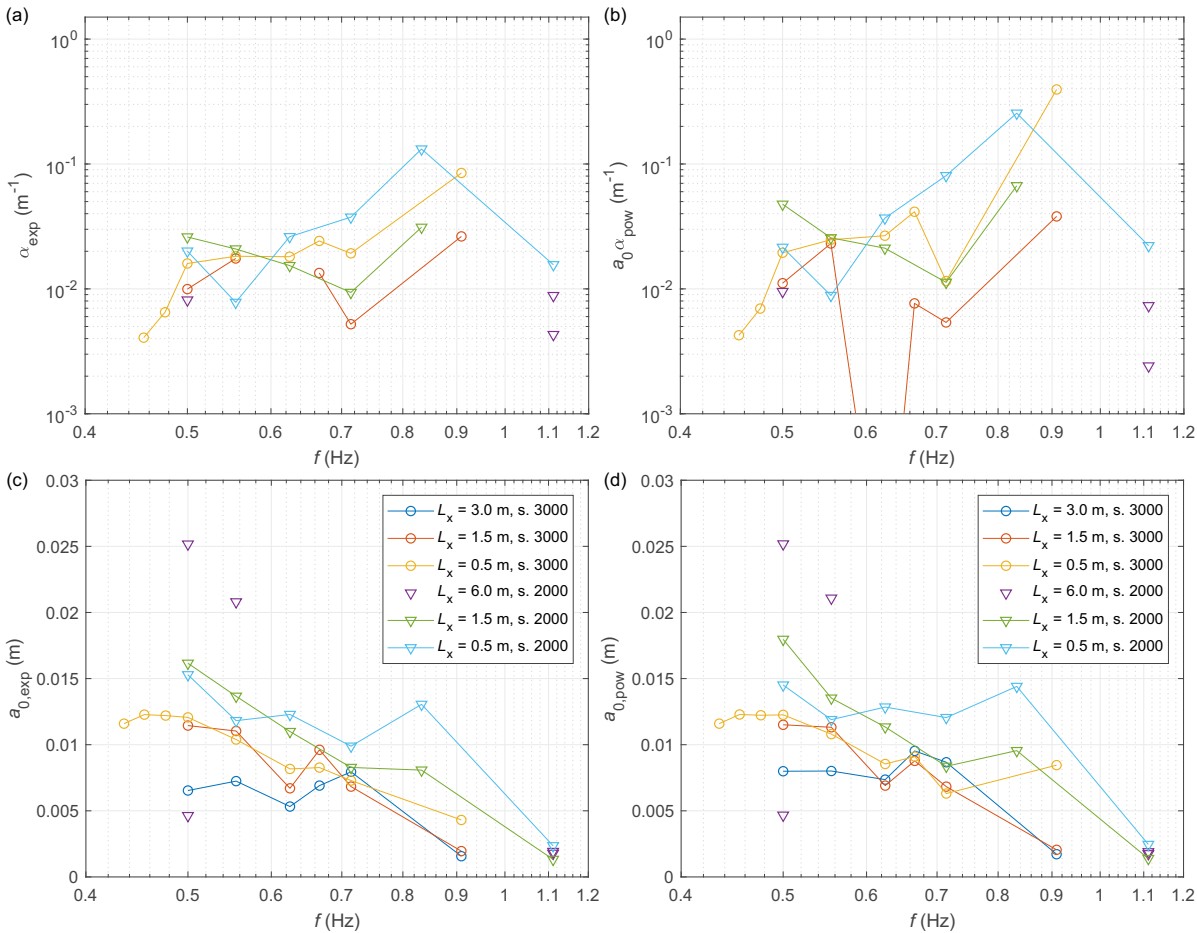

**Figure 5.** Attenuation rates (m$^{-1}$; a,b) and $a_0$ (m; c,d) in function of wave frequency $f$ (Hz) obtained by fitting function (3) (a,c) and (4) (b,d) to the observed LS-WICE data, with $a_0$ treated as a fitted coefficient. Note that in some tests (e.g., series 3000 with $L_x = 3.0$ m) the fitted values of $\alpha$ are negative and therefore are not shown in panels (a,b).

it is also worth noting that in some tests fitting the data with a constant produces fits with better quality than either function (3) or (4)). Accordingly, the fitted $a_0$ is in most cases much lower than both $a_{0,w}$ and $a_{\mathrm{MEEM,tot}}$, and even than the transmitted component $a_{\mathrm{MEEM},T_0}$; in some tests (e.g., 3160 or 3360) the difference exceeds 50%. Notably also, in a few tests the fitted attenuation rates are negative.

5    Conceptually speaking, because the attenuation takes place within the ice cover, the wave amplitude at $x_0$ should be that of the actual wave that has transmitted into the leading ice edge, which might be very different from $a_{0,w}$. However, in the case of LS-WICE, due to limited reflection at the ice edge, fixing the values of $a_0$ at the respective $a_{0,w}$ produces very regular results (Fig. 6), with consistent variability of $\alpha(f)$, even though, understandably, the goodness-of-fit measures become slightly worse (one free parameter instead of two). Several of the curves in Fig. 6 can be approximated with $\alpha \sim f^{n_\alpha}$ (for both attenuation





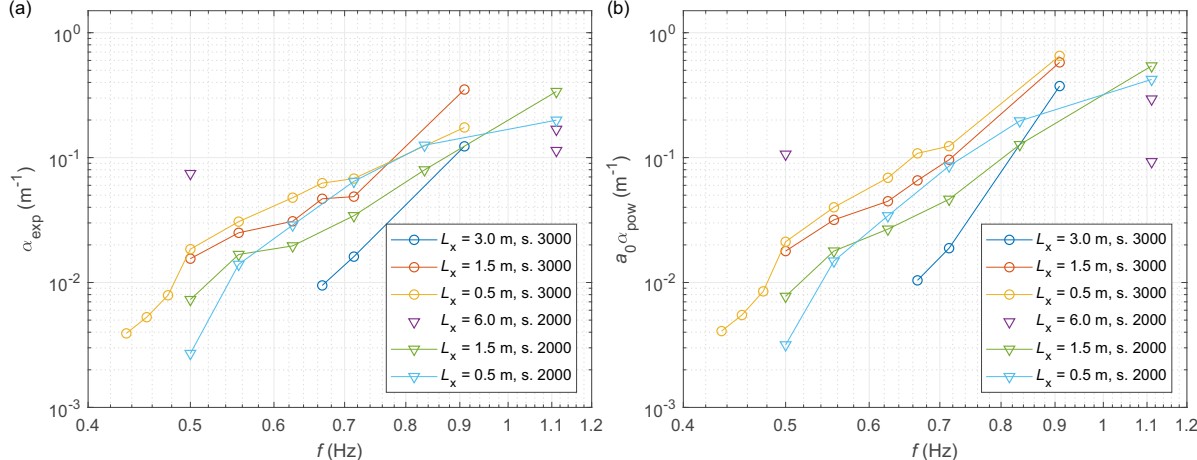

**Figure 6.** Attenuation rates ($\mathrm{m}^{-1}$) in function of wave frequency $f$ (Hz) obtained by fitting function (3) (a) and (4) (b) to the observed LS-WICE data, with $a_0$ fixed at $a_{0,w}$.

models), but the power $n_\alpha$ is extremely sensitive to individual data points (for example, removal of a single point, corresponding to the highest wave frequency, from the data from series 3200 changes $n_\alpha$ from 7 to 3.1), making any quantitative inferences regarding $\alpha(f)$ of little value.

The very different results obtained with fixed and fitted $a_0$ can be to a large degree attributed to the fact that no data points are available within the first 5 m from the ice edge. The attenuation within the region where the sensors were located was small, but the amplitudes in that region were, on average, much lower than the forcing amplitude $a_{0,w}$. Those differences, as the MEEM results indicate, cannot be explained by a reflection of wave energy from the ice edge. Thus, unless significant reflection took place at the floating boom, obviously not taken into account by the MEEM model, very strong attenuation must have taken place within the narrow zone between the ice edge and the first pressure sensors. We return to this issue further

in section 5, in the context of the results of numerical simulations. Here, it is worth stressing that – whatever the reasons for large amplitude differences between the wavemaker and the inner parts of the ice sheet – the fits with adjustable $a_0$s should be treated as suitable for that inner zone only, and should not be extrapolated to the region close to the ice edge. It should be also remembered that, due to limited number of data points and high spatial variability of wave amplitude related to the bending of the floes, discussed above, the fitted values of $\alpha_{\mathrm{exp}}$ and $\alpha_{\mathrm{pow}}$ are extremely sensitive to individual data points. On the other

hand, the fits with fixed $a_0$ by construction better represent relationships between the forcing wave amplitude and that within the ice, at the cost of less good agreement with the data within the inner zone of low attenuation.

Finally, it is worth noting that the available data do not provide arguments in favour of any of the two fit types considered. Especially when $a_0$ is treated as an adjustable parameter, the two fitted curves, (3) and (4), are in many tests almost undistinguishable (Supplementary Figs. 1 and 2). Although one function might better represent the data than the other for a particular test, the goodness-of-fit measures calculated globally for all tests are almost identical for both fitting functions. For example,

in the version with fitted $a_0$, the standard deviation of differences between the observed and fitted amplitudes (scaled with the




respective $a_{0,w}$) equals 0.0618 for function (3) and 0.0624 for function (4). When $a_0$ is fixed, the corresponding values are 0.0875 and 0.0714, respectively. The bias is in all cases below $10^{-2}$, negative with fitted $a_0$ and positive with fixed $a_0$. In spite of those similarities, however, with fixed $a_0$s, the fit (4) predicts stronger attenuation close to the ice edge and weaker attenuation further downwave (where, for small wave periods, e.g., in tests 3160, 3260, 3360, the exponential fit gives displacements

close to zero).

## 5   DEM simulation of the LS-WICE tests

In Part A, the coupled DEM–wave model was set up with sea ice properties corresponding to those from LS-WICE series 3000, and the model sensitivity was analyzed within a multi-dimensional parameter space. Here, we run the model for each test from series 3000, with floe length $L_x$, wave period $T$, and incident wave amplitude $a_{0,w}$ used in that test (Table 1). The

average floe–floe distance $d_f$ is very difficult to determine accurately. We fixed $d_f$ at $5 \cdot 10^{-3}$ m, one from the set of values tested in Part A that we find to be appropriate based on visual observations of the ice; undoubtedly, $d_f$ belongs to the model parameters with high uncertainty. As the group velocity is not available from observations, but necessary to solve the wave energy transport equation, the open water dispersion relation was assumed suitable for tests with $L_x = 0.5$ m and the elastic plate dispersion relation for tests with longer floes (see Fig. 2; note that significant differences between $c_g$ computed from those

two models are present only for the two higest wave frequencies considered).

With the basic settings listed above, the model was run several times for each test, with different combinations of the following four parameters: drag coefficient $C_{sd}$, restitution coefficient $\varepsilon$, and the coefficients $s_{min}$ and $c_{ow}$ used in (1) to calculate the overwash thickness. The goal of the simulations was to find combinations of those parameters that optimally reproduce the observational data over the available range of floe sizes and wave frequencies. This is equivalent with an assumption that

– as the general conditions and ice properties were stable throughout the whole test series 3000 – the same set of parameters should be suitable for reproducing wave attenuation in all tests. In other words, in the absence of evidence to the contrary, we disregard the possible dependence of $C_{sd}$, $\varepsilon$, etc. on wave period and amplitude.

Before we proceed to the analysis of the DEM simulations, it is worthwhile to notice that the suitable values of $C_{sd}$ can be constrained from observational data and from the results of the theoretical analysis in Part A. Due to the lack of observational

points close to the ice edge, it is reasonable to assume that the measurements represent the inner "regime", for which the attenuation rates $\alpha_{pow}$ are related to $C_{sd}$ by equation (2). As already discussed in the previous section, the values $\alpha_{pow}$ obtained by fitting with variable $a_0$ (more suitable for the inner zone than fitting with fixed $a_0$) vary strongly from test to test; however, for test series 3300 (yellow line in Fig. 5b), $\alpha_{pow}$ can be robustly fitted with expression (2), assuming open water dispersion relation and deep water, i.e., $\alpha = 8C_{sd}\omega^4/(3\pi g^2)$. The resulting value of $C_{sd}$ is very high and equals 4.04. More

generally, in order to obtain attenuation rates in the observed range $10^{-2}$–$10^{-1}$ m$^{-1}$ within the frequency range considered, values of $C_{sd}$ between $10^{-2}$ and $10^1$ are necessary. We leave the discussion of how realistic those values are for the next section. Here, we concentrate on analyzing those combinations of $C_{sd}$, $\varepsilon$, $s_{min}$ and $c_{ow}$ that produce results in good agreement with observational data. Two quantities are used as a measure of the model performance: standard deviation of differences $\sigma_{std}$





**Table 2.** Adjustable parameters of the model setups discussed in section 5 and the corresponding statistical measures of the model performance.

| Run | $C_{sd}$ | $\varepsilon$ | $s_{min}$ | $c_{ow}$ | $\sigma_{std}$ | $\delta_m$ |
|-----|----------|---------------|-----------|----------|----------------|------------|
| ID  | (–)      | (–)           | (–)       | (m)      | (–)            | (–)        |
| A   | 4.0      | 0.2           | 0.0000    | 0.0      | 0.085          | 0.080      |
| B   | 4.0      | 0.2–0.4       | 0.0150    | 0.3      | 0.083          | 0.079      |
| C   | 0.4      | 0.8           | 0.0150    | 0.4      | 0.083          | 0.077      |
| D   | 0.4      | 0.8           | 0.0125    | 0.4      | 0.082          | 0.075      |

In the case of setup B, two values of $\varepsilon$, 0.2 and 0.4, gave almost identical results in terms of both $\sigma_{std}$ and $\delta_m$.

and mean difference (bias) $\delta_m$ between the modelled and observed normalized amplitudes from 14 tests: 3210–3240, 3260, and all 9 tests from series 3300. Tests 3250 and 3140–3160 are included in the computations, but excluded from validation due to very strong influence of scattering-induced attenuation (3250) and due to problems with freezing and ice breaking (series 3100; see section 3). The model setups considered as satisfactory according to the above criteria, numbered from A to D, are summarized in Table 2. Figure 7 shows the simulated amplitude profiles along the tank for two selected tests; analogous plots for all tests can be found in Supplementary Fig. 4. Overall, in setups with no or weak overwash (i.e., high $s_{min}$ and low $c_{ow}$) both bias and standard deviation of differences increases with increasing $\varepsilon$, making only setups with the lowest $\varepsilon$ tested (0.2) satisfactory. On the other hand, setups with strong overwash and smaller ice–water drag show an opposite behavior, with relatively stable $\sigma_{std}$ for all values of $\varepsilon$, but with bias decreasing with increasing $\varepsilon$.

As expected from the analysis above, simulations without overwash require very high values of $C_{sd}$, and the optimal configuration is obtained for $C_{sd} = 4.0$ (the neighboring values tested equaled 3.0 and 5.0) and $\varepsilon = 0.2$ (setup A). Very similar values of $\sigma_{std}$ and $\delta_m$ are obtained with $c_{ow} = 0.4$ (setup B) – interestingly, the results of runs A and B are very close in terms of wave amplitudes in the region where the sensors were located, they mostly differ from each other in the area close to the ice edge in tests with short, and thus steep waves (e.g., test 3360), in which overwash contributes significantly to attenuation within the first few meters of the ice. This means that in the two setups, A and B, different processes contribute to wave dissipation at the ice edge, but their net effect further downwave is similar: dissipation due to overwash in B leads to a faster initial decrease of wave amplitudes, but the zone of strong dissipation becomes narrower. Notably, as far as we can judge from a visual analysis of the video material, setup B produces realistic results in terms of the extent of the overwash regions for different wave periods.

The extent of overwash is, as expected, larger in runs C and D, and in some tests the model predicts overwash over the entire ice sheet – which was not observed, but, before labeling those setups as unrealistic, it is worth noticing that the simulated dissipation due to overwash was in those cases extremely small over most of the area far from the ice edge. As can be seen from Fig. 7 and Supplementary Fig. 4, in spite of very similar $\sigma_{std}$ and $\delta_m$, the slope of the wave amplitude curves in setups C and D is very different from that in setups A and B – and, although $C_{sd}$ is in C and D an order of magnitude smaller than in A and B, the slope is higher in the first case. Whereas setups A and B tend to underestimate wave amplitudes at the ice edge





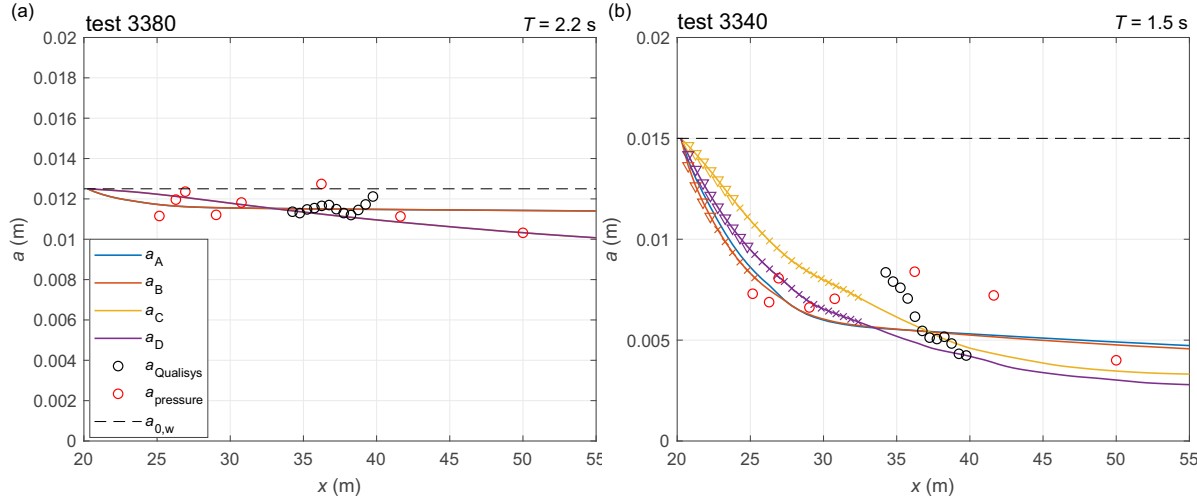

**Figure 7.** Amplitude of the vertical ice deflection along the tank in two selected tests from series 3000: 3380 (a) and 3340 (b; for analogous plots for all tests from series 3000, see Supplementary Fig. 4). The measured amplitude is shown with red (pressure sensors) and black (Qualisys) circles. The black dashed lines show the wavemaker amplitude $a_{0,w}$. The remaining lines show DEM results obtained with setups A–D (see Table 2). Locations with overwash are marked with crosses, and those where the change of wave amplitude due to overwash from one floe to the next exceeds 1 mm – with triangles.

and to overestimate them in the downwave region, setups C and D exhibit an opposite tendency. As said, this results in similar overall performance, although, obviously, some setups perform clearly better than others for individual tests.

In summary, it should be stressed that the values of $\sigma_{\mathrm{std}}$ in all four setups are comparable with analogous statistics for the least-square fits described in the previous section – in spite of the fact that parameters of each fit were optimized to the
5 individual test. In view of that, it is remarkable how well the same set of model parameters reproduces the observed variability of wave amplitudes over all tests considered.

## 6   Discussion and conclusions

The LS-WICE results analyzed in this work provide a very good example of how difficult it is to quantitatively assess wave energy attenuation in sea ice (especially in laboratory conditions, with a limited number of floes and over short distances)
and to attribute the observed attenuation to individual physical processes, even in a highly idealized laboratory setting. In spite of the simple geometry, regular wave forcing with small wave amplitudes, and highly uniform ice properties, several processes simultaneously modify wave propagation and dissipation, including floe collisions, floe breaking, overwash of the ice surface, production od slush, freezing between neighboring floes and between the ice and tank walls, and possibly some other. The results of observations and of the MEEM model clearly show that the scattering model alone does not explain the
observed spatial variability of wave amplitudes in fragmented ice, as the attenuation simulated with MEEM is, in most tests,



very low, much lower than the observed one. Another general conclusion drawn from the data analysis is that the attenuation rates increase with increasing wave frequency. The two facts together mean that the patterns of wave attenuation observed in LS-WICE are predominantly shaped by dissipative processes, and that the effectiveness of those processes in attenuating wave energy is frequency dependent.

An important aspect of the numerical part of this study is that several different combinations of the model parameters lead to reasonable agreement with observational data – even though we limited the number of adjustable parameters to three. It is very likely that even more regions of good model performance could be found within a higher-dimensional parameter space. Obviously, this ambiguity is a consequence of a large number of poorly constrained coefficients, large uncertainties in measured data, and the fact that the vertical deflection of the ice, being a combined effect of many processes, is the only validated

quantity. Although some combinations of the parameters seem more "realistic" than others, it is hard to favor one setup against the other without additional data. In particular, very high values of the drag coefficient in our "successful" setups are a few orders of magnitude higher than those reported in the literature, which are rarely larger than $10^{-2}$ (Castellani et al., 2018). The fact that the DEM reproduces the observed attenuation with so high drag coefficients indicates that, whatever were the dissipative processes actually contributing to attenuation, they can be described mathematically with formulae similar to those

used to compute skin drag in our model. In other words, it indicates that forces contributing to dissipation in LS-WICE were approximately proportional to the under-ice orbital velocities squared, and $C_{\mathrm{sd}}$ can be treated as an effective drag coefficient rather than a skin drag coefficient. Notably, very similar formulae to those underlying our model were used in a recent study by Voermans et al. (2019) to quantify turbulent kinetic energy dissipation rates under sea ice, with a result suggesting a rapid increase of the effective drag with increasing ice concentration. Although one should be extremely careful with extrapolating

data from lower ice concentrations (up to 0.8 in Voermans et al., 2019) to compact ice (close to 1 in our study), effective drag in the order of 1 seems plausible. In any case, in all our simulations with reasonable results, the necessary values of $C_{\mathrm{sd}}$ were higher than 0.1, unless extremely low $s_{\mathrm{min}}$ and high $c_{\mathrm{ow}}$ were used, producing strong overwash over much of the ice surface even in tests with long waves – which was not what was observed in the laboratory.

Undoubtedly, from the point of view of analyzing wave attenuation, a number of shortcomings can be listed in the LS-

WICE setup, and, based on this study, several recommendations can be formulated for future laboratory experiments designed specifically for measuring wave attenuation in fragmented sea ice (and, more generally, in other types of floating ice as well). First of all, it is crucial to locate sensors measuring the vertical deflection of the ice, acceleration of ice floes, and possibly other quantities, in the zone of the strongest attenuation close to the ice edge. At the same time, it is important not to limit the observations to that zone, as the attenuation further downwave is very likely much weaker and cannot be extrapolated

from that observed at the ice edge. Moreover, distributing sensors over a possibly large distance is necessary if the suitability of alternative theoretical attenuation curves is to be tested. The LS-WICE data, as demonstrated in this work, are clearly not sufficient for that purpose. It must be remembered, however, that this requirement is easy to formulate, but very hard to fulfill in a wave tank, as its length limits the possible number of wavelengths. Further, as discussed at the beginning of this section, LS-WICE shows that, although it would be desirable to design experiments eliminating all other dissipation mechanisms except

the one of interest, this goal is very hard, if not impossible to achieve. Some factors present in LS-WICE, e.g., freezing to the





side walls, can be eliminated with some effort (though it is not straightforward in an ice tank several tens of meters long), but the influence of other factors has to be accepted and, as they are impossible to eliminate, quantified. In particular, overwash is very difficult to eliminate in laboratory conditions due to small thickness and therefore very low freeboard of the ice, so that, apart from recording overwash presence and extent by video, assessment of its thickness is desirable, enabling formulation of
more advanced parameterizations than the primitive one proposed in this study.

Finally, as already mentioned in the discussion in Part A, integrating scattering effects in the DEM model presented here is a major challenge that must be addressed to make the model suitable for analyzing mutual relationships between non-dissipative and dissipative processes contributing to wave energy attenuation. Obviously, attenuation in real sea ice is not a simple superposition of individual processes that can be considered independently of each other, as in the present study.

*Code and data availability.*   The code of the DESIgn model is freely available at https://herman.ocean.ug.edu.pl/LIGGGHTSseaice.html and as supplementary material to Herman (2016). The extended code necessary to reproduce the results presented in this paper, together with input scripts, can be obtained from the corresponding author. The LS-WICE data used in this paper are described in the data storage report available at https://zenodo.org/record/1067170 and can be obtained from the authors.

*Author contributions.*   All authors contributed to planning of the research and to discussion and analysis of the results. S.C. performed the
analysis of experimental data and the MEEM simulations. A.H. performed the numerical simulations and wrote the text.

*Competing interests.*   The authors declare no competing interests

*Acknowledgements.*   The development of the numerical model used in this work has been financed by the Polish National Science Centre research grant No. 2015/19/B/ST10/01568 ("Discrete-element sea ice modeling – development of theoretical and numerical methods"). Coauthors SC and HHS are supported in part by ONR grant No. N00014-17-1-2862. The laboratory work described in this publication was
supported by the European Community's Horizon 2020 Programme through the grant to the budget of the Integrated Infrastructure Initiative HYDRALAB+, Contract no. 654110. The authors would like to thank the Hamburg Ship Model Basin (HSVA), especially the ice tank crew, for the hospitality, technical and scientific support and the professional execution of the test programme in the Research Infrastructure ARCTECLAB.



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
