# Peer review of "Wave energy attenuation in fields of colliding ice floes. Part B: A laboratory case study"

_The Cryosphere, 2019_

## Referee Comment (RC1) · Anonymous Referee #1 · 18 Jul 2019

**1   General comments**

This paper is about the attenuation of ocean waves as they pass through fields of sea ice. It is the second instalment of a two-part connected study, which follows a Part A focussed predominantly on modelling. I have been asked to review both papers. As expressed in my review of Part A, a single longer paper combining Parts A and B would certainly have made my job easier because of intersections between the two parts. Be that as it may, I focus here on Part B which reports on a set of experiments done in a wave flume interpreted using theory from Part A and the Kohout et al. (2007) model.

I support publication of Part B (and Part A, incidentally), subject to minor amendments. As with my review of Part A, many of my specific comments below are intended to

clarify and hopefully improve the manuscript.

The laboratory experiments analyzed in this paper were performed in a large ice model basin at the Hamburg Ship Model Basin (HSVA). Initial analyses of these data with somewhat different orientations have already been completed and published by the authors and others. In essence, the purpose of the experiments was to replicate how long-crested waves interact with densely packed ice floes at the scale of a wave flume, the HSVA facility providing a unique infrastructure to achieve this expeditiously by allowing parallel experiments to be done contemporaneously because of its width. All experiments and the models used to analyze the data are inherently one-dimensional.

Somewhat predictably, the authors find that the Kohout et al. (2007) model (designated MEEM after the mathematical method used), which computes the conservative redistribution of wave energy arising from successive reflections at the edges of consecutive ice floes, cannot replicate the attenuation observed in the experimental data, but the discrete-element model (DEM) reported in the companion Part A paper can. However, as the authors state "more than one combination of the parameters of the coupled DEM–wave model (restitution coefficient, drag coefficient, overwash criteria) produces spatial attenuation patterns in good agreement with observed ones over a range of wave periods and floe sizes, making selection of optimal model settings difficult. The results demonstrate that experiments aimed at identifying dissipative processes accompanying wave propagation in sea ice and quantifying the contribution of those processes to the overall attenuation require simultaneous measurements of many processes over possibly large spatial domains." This is a crucial revelation, as the Part A DEM is necessarily limited in its quantification of the physical processes that act to dissipate wave energy—which are many and hard to model. It intimates a rather different, more empirical, statistical approach to simulating the effects of sea ice on ocean waves in the future, especially for wave forecasting and large scale modelling predictions where compromised sea ice covers and the uptrending of global winds and wave height consequent to a warming climate are now paramount. If, as the authors

Interactive
comment

point out "the main goal of [their] study is to demonstrate that the interpretation of the observed attenuation and validation of numerical models based on that type of data is problematic, as many mutually interrelated mechanisms contribute to the net attenuation," one has to ask whether there is any value at all in trying to parametrize the underlying physics or is it simply wiser to just work with features in the data themselves?

The experimental programme is impressive with respect to its thoroughness, covering a good range of floe lengths and wave periods commensurate with the size of the flume, with sophisticated measurement sensors including video to clarify features in the behavioural response of floes. Floes are always rectangular, which is appropriate given the one-dimensionality of the experiments. Overwashing of the floes is also considered. Relevance to Nature, i.e. whether results can be scaled up to wave-ice interactions in naturally occurring sea ice fields, goes notably unreported and this is really my only weighty criticism mentioned below under Specific Comments.

The Part A DEM predicts two interesting outcomes that I mention specifically here: nonexponential attenuation that depends strongly on the group velocity via the dispersion relation and, consequently, on the ice morphology; and wave amplitude profiles that identify the existence of two zones—a narrow zone near the ice edge where energetic collisions lead to high attenuation rates, and an interior zone with densely packed ice floes where attenuation is less. As foreshadowed above, I find the outcome that the MEEM model cannot reproduce the attenuation seen in the wave flume less interesting as I would not have expected a conservative model, based entirely on reflections occurring due to impedance changes at what are a relatively modest number of ice cakes in a wave flume, to predict much attenuation at all.

**2   Specific comments**

1. The authors do not mention at all how to scale up their results to ocean wave-lengths and sea ice floes in Nature. Is Froude scaling appropriate? Some discussion on scaling should be included in the final paper to move it from an interesting analysis of some laboratory data to work that has implications for the Earth's polar and subpolar seas. Specifically, can the authors assure the reader that Froude scaling is adequate to extrapolate model scale tank testing to full scale sea ice covers such as the marginal ice zone. While this sounds daunting, can I suggest that the DEM might be used to do this.

2. Can I encourage the authors to say something about the ramifications of their work to how the topic of wave-ice interactions might evolve in the future, given their strong statements about fitting models to data. If the authors agree with the (uncitable) comments I have made in my review, they should discuss in their paper what this implies with respect to (i) how the effects of ocean waves should be embedded in large scale sea ice models—as the way floes break up and move about depends in part on attenuation; and (ii) how wave forecasting models should parametrize sea ice covers with differing morphologies to provide realistic wave decay rates. As the authors write "many different models can be calibrated to reproduce observations with reasonable accuracy, especially considering large uncertainties in attenuation rates derived from measurements."

3. Unlike Part A, Part B includes a (finite) set of each of the possible modes in the MEEM theory. The explanation of the model is brief but, given it has been published before (see Kohout et al. 2007 and related papers), it is sufficient. What I am unsure about is how the model is applied in the context of the wave flume, which has a wave-maker, a forward and terminal ice edge, and a beach. Can the authors please explain this in more detail.

4. §2.2 is a summary of the DEM model, described more exhaustively in Part A. (I have already said that I would have preferred a single paper, so I am not going to labour this point here.)

5. The section on laboratory observations is well written and thorough, with Fig. 1 providing a good schematic overview of the set up and Table 1 listing the experimental parameters themselves. It is evident that the experiments were done with care and that data were rejected when unforeseen mishaps occurred. Nonetheless, I always seek reassurance that the beach is effective for these types of experiment as, without active control, returning wave fronts can rather mess up an otherwise well planned experimental design. Can the authors please provide percentage beach efficiencies as a function of wave period and also reassure me that no ice was present over the beach?

6. I was pleased to hear that strong overwash was limited to the zone near the frontmost ice edge and that it was mostly associated with steep waves. This is what I would expect, as the strong attenuation that is being observed in the outer zone would soon reduce the waves sufficiently to minimize overwash farther in, even with the small freeboard of the ice in the experiment.

7. The authors explain features of the dispersion in Fig. 2 convincingly.

8. Fig. 3 shows how the wave amplitude $a(x)$ varies with distance $x$ measured down the wave flume, including data, MEEM results and least square fits to equations from Part A reproduced in Part B as equations (3) and (4) with attenuation coefficients $\alpha_{\mathrm{exp}}$ and $\alpha_{\mathrm{pow}}$ respectively. The lack of fit to MEEM is very apparent but I am not surprised by this because of the number of floes involved—even when $L_x = 0.5$ m, and their sizes relative to wavelength. A *resonance effect* seems to occur in companion tests when wavelengths are roughly twice the floe size; again this effect has been documented in other publications, e.g. those of Meylan. As hinted above, a flawless application of the Kohout et al. theory for ice

cakes in a physical wave flume, where there is also likely to be considerable dissipation due to several mostly-nonlinear processes, is also challenging, making MEEM an unworthy candidate to fit such experimental data. Incidentally, can I assume in Fig. 4 because there is open water on each side of the ice cover, that $|a_{\mathrm{MEEM},R_0}|^2 + |a_{\mathrm{MEEM},T_0}|^2 = 1$ for plots (b) and (c)? If so, say this, and the figure could be simplified.

9. In applying the DEM to a good proportion of the data to select the *best* $C_{\mathrm{sd}}$, $\varepsilon$, $s_{\mathrm{min}}$ and $c_{\mathrm{ow}}$ using $\sigma_{\mathrm{std}}$ and $\delta_{\mathrm{m}}$ (see Part B page 14–15 text), the authors wisely acknowledge the uncertainty around some of the parameters required to drive the model, e.g. floe separation. And, not knowing the actual dispersion relation, they have had to make an educated decision to get the group velocity $c_g$. Table 2 shows 4 Runs A–D meeting documented satisfactory quality criteria; see also Fig. 7. The supporting discussion is very detailed and sometimes a little confusing, unfortunately.

10. For example, starting at line 10 on page 15 we read that "simulations without overwash require very high values of $C_{\mathrm{sd}}$ and the optimal configuration is obtained for $C_{\mathrm{sd}} = 4.0$." OK, got that. Then we read that "runs A and B . . . mostly differ from each other in the area close to the ice edge in tests with short, and thus steep waves (e.g., text 3360) in which overwash contributes significantly . . . " But I thought overwash was minimal in Runs A and B. Having said this, the premise is fine, namely that different processes can produce similar net effects. I recommend page 15 and the first half of page 16 are rewritten to be clearer; even if I am misunderstanding, my example suggests the current prose is ambiguous.

11. I agree that the authors have shown "how difficult it is to quantitatively assess wave energy attenuation in sea ice (especially in laboratory conditions, with a limited number of floes and over short distances) and [crucially] to attribute the observed attenuation to individual physical processes, even in a highly idealized

laboratory setting." As I have already said, this is a tremendously important outcome of this work. Plagiarizing the authors' words, ... more generally this is a consequence of a large number of poorly constrained coefficients, large uncertainties in measured data, and the fact that the vertical deflection of the ice, being a combined effect of many processes, is invariably the only validated quantity.

12. I reiterate that I am less excited by the fact that scattering alone doesn't explain the observations, as I would not have expected it to. But I also remind the authors that (i) they have used a one-dimensional model and one-dimensional data; and (ii) they have not considered that in a real ice field there will be dissipation of the scattered waves in the regions between ice floes whether these are ice-free or filled with brash.

13. But I do agree with the authors statement that "The patterns of wave attenuation observed in [the wave flume tests] are predominantly shaped by dissipative processes, and that the effectiveness of those processes in attenuating wave energy is frequency dependent."

14. While the arguments around using $C_{\mathrm{sd}}$ as an effective drag coefficient are convincing and recalling that the authors do actually mention what happens when a linear drag law is used instead of a quadratic one in Part A (viz. that exponential decay as opposed to power law decay eventuates), in general, i.e. for real sea ice fields, this idea would need to be carefully validated. It is tantamount to packaging a smorgasbord of dissipative mechanisms under the quadratic drag law umbrella. Certainly nice if you can do it!

**3  Technical corrections**

1. Page 1, line 20. The Part A paper is not in the bibliography. In fact I had to source it from ResearchGate.

2. Page 15, line 12. Should $c_{\mathrm{OW}} = 0.3$ here to match Table 2? The wording is such that I am unsure whether you are comparing with the previous sentence, i.e. Run A, or are saying $c_{\mathrm{OW}} = 0.4$ gives the same result as in Table 2. Either way, it is ambiguous so should be clarified.

---

## Referee Comment (RC2) · Anonymous Referee #2 · 5 Aug 2019

The paper describes laboratory measurements of wave dispersion and attenuation in sea ice collected in the Hamburg Ship Model Basin (HSVA). It is connected to another paper concerned with theoretical modeling finalized to a theoretical representation of the data reported. The experimental set-up and the methods used to perform measurements are well described and denote the huge authors' expertise in this field. Observed wave attenuation data were compared with two models. The first was a scattering model by an ensemble of elastic floating plates using a method based on a matched eigenfunction expansion (MEEM); the second was a dissipative model of sea ice based on a discrete-element model (DEM), which is detailed in the companion paper (Part A). Results show that the MEEM model predicts lower attenuations than those measured, thus revealing the chief contribution of dissipative mechanisms as described in DEM

to the observed attenuations. Attenuation was frequency-dependent, but a non-unique set of the tuning parameters (3 in total) was found to explain the observations. Although authors deeply discuss this behavior in order to find the causes, i.e. the uncertainties of the measured data, they did not arrive at a unique conclusion for which a unique set of fitting parameters could be fixed. It resulted that both exponential and power attenuation law explained the collected data with comparable accuracy. This outcome reveals how hard would be to ascertain the nature of the dissipative processes occurring the MIZ.

I support this paper. I have only minor comments for the authors: 1) in order to cast the presented results into a realistic frame, I would be happy the authors agree to add a discussion about the scaling to the ocean wavelengths and sea ice sizes occurring in the Polar regions. 2) Figure 2 is important but not well explained. Please add further comments with special focus on the assumed wave dispersion relationship.

---

## Author Comment (AC1) · 12 Sep 2019

**Response to the comments of Reviewer #1**

This paper is about the attenuation of ocean waves as they pass through fields of sea ice. It is the second instalment of a two-part connected study, which follows a Part A focussed predominantly on modelling. I have been asked to review both papers. As expressed in my review of Part A, a single longer paper combining Parts A and B would certainly have made my job easier because of intersections between the two parts. Be that as it may, I focus here on Part B which reports on a set of experiments done in a wave flume interpreted using theory from Part A and the Kohout et al. (2007) model.

I support publication of Part B (and Part A, incidentally), subject to minor amendments. As with my review of Part A, many of my specific comments below are intended to clarify and hopefully improve the manuscript.

The laboratory experiments analyzed in this paper were performed in a large ice model basin at the Hamburg Ship Model Basin (HSVA). Initial analyses of these data with somewhat different orientations have already been completed and published by the authors and others. In essence, the purpose of the experiments was to replicate how long-crested waves interact with densely packed ice floes at the scale of a wave flume, the HSVA facility providing a unique infrastructure to achieve this expeditiously by allowing parallel experiments to be done contemporaneously because of its width. All experiments and the models used to analyze the data are inherently one-dimensional.

Somewhat predictably, the authors find that the Kohout et al. (2007) model (designated MEEM after the mathematical method used), which computes the conservative redistribution of wave energy arising from successive reflections at the edges of consecutive ice floes, cannot replicate the attenuation observed in the experimental data, but the discrete-element model (DEM) reported in the companion Part A paper can.

However, as the authors state "more than one combination of the parameters of the coupled DEM– wave model (restitution coefficient, drag coefficient, overwash criteria) produces spatial attenuation patterns in good agreement with observed ones over a range of wave periods and floe sizes, making selection of optimal model settings difficult. The results demonstrate that experiments aimed at identifying dissipative processes accompanying wave propagation in sea ice and quantifying the contribution of those processes to the overall attenuation require simultaneous measurements of many processes over possibly large spatial domains." This is a crucial revelation, as the Part A DEM is necessarily limited in its quantification of the physical processes that act to dissipate wave energy— which are many and hard to model. It intimates a rather different, more empirical, statistical approach to simulating the effects of sea ice on ocean waves in the future, especially for wave forecasting and large scale modelling predictions where compromised sea ice covers and the uptrending of global winds and wave height consequent to a warming climate are now paramount. If, as the authors point out "the main goal of [their] study is to demonstrate that the interpretation of the observed attenuation and validation of numerical models based on that type of data is problematic, as many mutually interrelated mechanisms contribute to the net attenuation," one has to ask whether there is any value at all in trying to parametrize the underlying physics or is it simply wiser to just work with features in the data themselves?

As we already wrote in our reply to the comments regarding part A:
1. We do not think this result is a "revelation". It is a fact that is often disregarded/forgotten, or at least not sufficiently acknowledged, but not very unexpected. The observational data (both from the lab and from the field) can be fitted with different functions with similar accuracy, and thus it

should not be surprising that also different model formulations result in similar agreement with that data.

2. The dilemma whether data-driven models are better than those based on physics is obviously not limited to models of wave-ice interactions, and, in our view, the answer depends on the purpose of a given model. If computational efficiency and predictive skills are a priority (e.g. in models working operationally and used for forecasting), then data-driven, black box models are hard to beat. If, however, the purpose of a model is to improve our understanding of certain processes and interactions, we need both, theories and observations, because only combining the two can bring us forward. And, well, there is nothing new in those statements.
   In the particular case of wave attenuation in sea ice, our "revelation" is simply that, when one wants to infer about processes contributing to attenuation, measuring attenuation rates alone is not sufficient. We don't think this finding is as dramatic as the Reviewer suggests.

The experimental programme is impressive with respect to its thoroughness, covering a good range of floe lengths and wave periods commensurate with the size of the flume, with sophisticated measurement sensors including video to clarify features in the behavioural response of floes. Floes are always rectangular, which is appropriate given the one-dimensionality of the experiments. Overwashing of the floes is also considered. Relevance to Nature, i.e. whether results can be scaled up to wave-ice interactions in naturally occurring sea ice fields, goes notably unreported and this is really my only weighty criticism mentioned below under Specific Comments.

The Part A DEM predicts two interesting outcomes that I mention specifically here: nonexponential attenuation that depends strongly on the group velocity via the dispersion relation and, consequently, on the ice morphology; and wave amplitude profiles that identify the existence of two zones—a narrow zone near the ice edge where energetic collisions lead to high attenuation rates, and an interior zone with densely packed ice floes where attenuation is less. As foreshadowed above, I find the outcome that the MEEM model cannot reproduce the attenuation seen in the wave flume less interesting as I would not have expected a conservative model, based entirely on reflections occurring due to impedance changes at what are a relatively modest number of ice cakes in a wave flume, to predict much attenuation at all.

We were not surprised by the fact the MEEM did not reproduce the observed attenuation. Rather, we used the MEEM model to demonstrate what we'd expected and to estimate the contribution of scattering to the overall attenuation (once more: expected to be small). MEEM proved also very helpful in identifying those few tests in which scattering *did* have dominant influence on attenuation, so that we could eliminate them from DEM validation.

Specific comments

1. The authors do not mention at all how to scale up their results to ocean wavelengths and sea ice floes in Nature. Is Froude scaling appropriate? Some discussion on scaling should be included in the final paper to move it from an interesting analysis of some laboratory data to work that has implications for the Earth's polar and subpolar seas. Specifically, can the authors assure the reader that Froude scaling is adequate to extrapolate model scale tank testing to full scale sea ice covers such as the marginal ice zone. While this sounds daunting, can I suggest that the DEM might be used to do this.

   It is true that the present paper does not consider cases that cover the field conditions. It is not the

goal of this paper. Our study is a humble first step towards such a goal. Most importantly, the paper successfully builds a numerical model, demonstrating that a DEM is a good tool for studying wave-ice interaction problems, and points out the difficulty of isolating various mechanisms using only the dispersion/attenuation data. We strongly believe that this finding is applicable not only to idealized laboratory cases, but (presumably even more so) to field situations as well.

We do agree with the suggestion to apply our DEM model (after necessary extensions/modifications) to full scale sea ice problems – we do plan doing this in the future, but this goal is far beyond the scope of the present paper.

Self-citing our reply to a similar comment from Reviewer #2: "To be honest, we do not feel comfortable with extrapolating any quantitative data from the lab scale to the field scale. Our paper validated the DEM tool and revealed many important and fundamental aspects of wave-ice interaction. But the numbers obtained cannot be directly applied to field, and one should be extremely careful when attempting such scaling. To do so, a separate study with many more DEM runs will be needed, and we plan to use our present results as a starting point for more realistic (and larger-scale) computations in the future.

Nevertheless, as both Reviewers raise the issue of scaling in their comments, we will add some discussion on that to the revised paper."

2. Can I encourage the authors to say something about the ramifications of their work to how the topic of wave-ice interactions might evolve in the future, given their strong statements about fitting models to data. If the authors agree with the (uncitable) comments I have made in my review, they should discuss in their paper what this implies with respect to (i) how the effects of ocean waves should be embedded in large scale sea ice models—as the way floes break up and move about depends in part on attenuation; and (ii) how wave forecasting models should parametrize sea ice covers with differing morphologies to provide realistic wave decay rates. As the authors write "many different models can be calibrated to reproduce observations with reasonable accuracy, especially considering large uncertainties in attenuation rates derived from measurements."

As we already wrote in our reply to the previous comments: If the purpose of a given model is to provide reliable and fast predictions of wave attenuation rates given a certain combination of wave forcing and sea ice properties, then the accuracy and computational efficiency – and not the question whether the underlying physics is captured – are the most important criteria for selecting "the best" algorithm.

An aspect that was not yet mentioned in our previous comments: In fully coupled ocean/sea ice/atmosphere models, it is very important not only to predict accurate wave attenuation rates, but also to "assign" the dissipated energy to the correct "place" – it is not the same whether dissipation takes place in water due to, e.g., turbulence generation, or within the ice due to, e.g., inelastic or viscous processes. Thus, understanding the underlying physics and capturing it in those models should be very important. Once more, for that we need more measurements of many variables, not only wave attenuation rates.

3. Unlike Part A, Part B includes a (finite) set of each of the possible modes in the MEEM theory. The explanation of the model is brief but, given it has been published before (see Kohout et al. 2007 and related papers), it is sufficient. What I am unsure about is how the model is applied in the context of the wave flume, which has a wave-maker, a forward and terminal ice edge, and a beach. Can the authors please explain this in more detail.

When applying MEEM, we assumed that the open water in front of and behind the ice floes was

infinite, i.e., we used the model exactly as in the works by Kohout and colleagues. Notably, Kohout compared her MEEM modeling results to the tank experiments by Sakai and Hanai (2002), in which the overall setup was almost identical to that used in LS-WICE (wavemaker, wave absorber, etc.). A very important fact about our measurements (described in Cheng et al, 2018) is that the wave amplitudes along the tank were determined based on short time series, before the wave reflected from the beach could reach the sensors. Thus, no effects related to waves reflected from the end of the tank were present in our data.

4. § 2.2 is a summary of the DEM model, described more exhaustively in Part A. (I have already said that I would have preferred a single paper, so I am not going to labour this point here.)

   -

5. The section on laboratory observations is well written and thorough, with Fig. 1 providing a good schematic overview of the set up and Table 1 listing the experimental parameters themselves. It is evident that the experiments were done with care and that data were rejected when unforeseen mishaps occurred. Nonetheless, I always seek reassurance that the beach is effective for these types of experiment as, without active control, returning wave fronts can rather mess up an otherwise well planned experimental design. Can the authors please provide percentage beach efficiencies as a function of wave period and also reassure me that no ice was present over the beach?

   As we write in response to comment #3: Since only the part of the time series before the reflected wave arrived at the monitored section was used in the study, this is not an issue.

6. I was pleased to hear that strong overwash was limited to the zone near the frontmost ice edge and that it was mostly associated with steep waves. This is what I would expect, as the strong attenuation that is being observed in the outer zone would soon reduce the waves sufficiently to minimize overwash farther in, even with the small freeboard of the ice in the experiment.

   Yes, we were pleased by that observation as well. Definitely, tests with overwash over the entire ice sheet were not what we wanted to have!

7. The authors explain features of the dispersion in Fig. 2 convincingly.

   Thank you!

8. Fig. 3 shows how the wave amplitude a(x) varies with distance x measured down the wave flume, including data, MEEM results and least square fits to equations from Part A reproduced in Part B as equations (3) and (4) with attenuation coefficients $\alpha_{exp}$ and $\alpha_{pow}$ respectively. The lack of fit to MEEM is very apparent but I am not surprised by this because of the number of floes involved— even when $L_x = 0.5m$, and their sizes relative to wavelength. A resonance effect seems to occur in companion tests when wavelengths are roughly twice the floe size; again this effect has been documented in other publications, e.g. those of Meylan. As hinted above, a flawless application of the Kohout et al. theory for ice cakes in a physical wave flume, where there is also likely to be considerable dissipation due to several mostly-nonlinear processes, is also challenging, making MEEM an unworthy candidate to fit such experimental data. Incidentally, can I assume in Fig. 4 because there is open water on each side of the ice cover, that $|a_{MEEM;R_0}|^2 + |a_{MEEM;T_0}|^2 = 1$ for plots (b) and (c)? If so, say this, and the figure could be simplified.

As we wrote in one of our earlier replies, we were not surprised by the MEEM results. The repeating comments from the Reviewer saying that the MEEM results look exactly how they should be expected to look like indicates that we definitely should state this very clearly in our revised paper! As we write in our earlier reply, we used MEEM to estimate the influence of scattering and to identify the tests in which that influence was dominant (as suggested, we will add references to papers in which such resonance was described).

As for the last sentence: yes, the (normalized) squared amplitudes of the wave reflected from the ice edge and the propagating wave leaving the ice sheet at the end of the tank should sum up to 1. We agree that the figure can be simplified and we will prepare a simpler one for the revised paper.

9. In applying the DEM to a good proportion of the data to select the best Csd, ε, smin and cow using σstd and δm (see Part B page 14–15 text), the authors wisely acknowledge the uncertainty around some of the parameters required to drive the model, e.g. floe separation. And, not knowing the actual dispersion relation, they have had to make an educated decision to get the group velocity $c_g$. Table 2 shows 4 Runs A–D meeting documented satisfactory quality criteria; see also Fig. 7. The supporting discussion is very detailed and sometimes a little confusing, unfortunately.

We will try to rewrite this fragment of the text to make it less confusing.

10. For example, starting at line 10 on page 15 we read that "simulations without overwash require very high values of Csd and the optimal configuration is obtained for Csd = 4.0." OK, got that. Then we read that "runs A and B . . . mostly differ from each other in the area close to the ice edge in tests with short, and thus steep waves (e.g., text 3360) in which overwash contributes significantly . . . " But I thought overwash was minimal in Runs A and B. Having said this, the premise is fine, namely that different processes can produce similar net effects. I recommend page 15 and the first half of page 16 are rewritten to be clearer; even if I am misunderstanding, my example suggests the current prose is ambiguous.

As stated above, we will rewrite that fragment in the revised paper.

11. I agree that the authors have shown "how difficult it is to quantitatively assess wave energy attenuation in sea ice (especially in laboratory conditions, with a limited number of floes and over short distances) and [crucially] to attribute the observed attenuation to individual physical processes, even in a highly idealized laboratory setting." As I have already said, this is a tremendously important outcome of this work. Plagiarizing the authors' words, . . . more generally this is a consequence of a large number of poorly constrained coefficients, large uncertainties in measured data, and the fact that the vertical deflection of the ice, being a combined effect of many processes, is invariably the only validated quantity.

-

12. I reiterate that I am less excited by the fact that scattering alone doesn't explain the observations, as I would not have expected it to. But I also remind the authors that (i) they have used a one-dimensional model and one-dimensional data; and (ii) they have not considered that in a real ice field there will be dissipation of the scattered waves in the regions between ice floes whether these are ice-free or filled with brash.

We definitely have to clearly state in the revised paper that we did not expect the MEEM model to explain the observations! And yes, the situation might be very different at lower ice concentration and in a 2D setting.

13. But I do agree with the authors statement that "The patterns of wave attenuation observed in [the wave flume tests] are predominantly shaped by dissipative processes, and that the effectiveness of those processes in attenuating wave energy is frequency dependent."
-

14. While the arguments around using Csd as an effective drag coefficient are convincing and recalling that the authors do actually mention what happens when a linear drag law is used instead of a quadratic one in Part A (viz. that exponential decay as opposed to power law decay eventuates), in general, i.e. for real sea ice fields, this idea would need to be carefully validated. It is tantamount to packaging a smorgasbord of dissipative mechanisms under the quadratic drag law umbrella. Certainly nice if you can do it!

This is what practically everyone is doing in their models: the quadratic drag law is very widely used in models, and in some models the drag coefficient is used as a calibration coefficient.
In the context of our and similar studies, it is worth remembering that the choice of a particular drag law influences the functional form of the wave attenuation curves.

Technical corrections

1. Page 1, line 20. The Part A paper is not in the bibliography. In fact I had to source it from ResearchGate.

At the stage of submitting it was not clear how to cite A in B and vice versa. We will add the proper references to both bibliographies in the revised version.

2. Page 15, line 12. Should cow = 0.3 here to match Table 2? The wording is such that I am unsure whether you are comparing with the previous sentence, i.e. Run A, or are saying cow = 0.4 gives the same result as in Table 2. Either way, it is ambiguous so should be clarified.

Yes, thank you for pointing that out! We will correct it in the revised paper.

---

## Author Comment (AC2) · 12 Sep 2019

**Response to the comments of Reviewer #2**

The paper describes laboratory measurements of wave dispersion and attenuation in sea ice collected in the Hamburg Ship Model Basin (HSVA). It is connected to another paper concerned with theoretical modeling finalized to a theoretical representation of the data reported. The experimental set-up and the methods used to perform measurements are well described and denote the huge authors' expertise in this field. Observed wave attenuation data were compared with two models. The first was a scattering model by an ensemble of elastic floating plates using a method based on a matched eigenfunction expansion (MEEM); the second was a dissipative model of sea ice based on a discrete-element model (DEM), which is detailed in the companion paper (Part A). Results show that the MEEM model predicts lower attenuations than those measured, thus revealing the chief contribution of dissipative mechanisms as described in DEM to the observed attenuations. Attenuation was frequency-dependent, but a non-unique set of the tuning parameters (3 in total) was found to explain the observations. Although authors deeply discuss this behavior in order to find the causes, i.e. the uncertainties of the measured data, they did not arrive at a unique conclusion for which a unique set of fitting parameters could be fixed. It resulted that both exponential and power attenuation law explained the collected data with comparable accuracy. This outcome reveals how hard would be to ascertain the nature of the dissipative processes occurring the MIZ.

Yes, and as we write in reply to your second comment in part A, this is a very important and practically relevant conclusion from our study.

I support this paper. I have only minor comments for the authors: 1) in order to cast the presented results into a realistic frame, I would be happy the authors agree to add a discussion about the scaling to the ocean wavelengths and sea ice sizes occurring in the Polar regions. 2) Figure 2 is important but not well explained. Please add further comments with special focus on the assumed wave dispersion relationship.

1. To be honest, we do not feel comfortable with extrapolating any quantitative data from the lab scale to the field scale. Our paper validated the DEM tool and revealed many important and fundamental aspects of wave-ice interaction. But the numbers obtained cannot be directly applied to field, and one should be extremely careful when attempting such scaling. To do so, a separate study with many more DEM runs will be needed, and we plan to use our present results as a starting point for more realistic (and larger-scale) computations in the future. Nevertheless, as both Reviewers raise the issue of scaling in their comments, we will add some discussion on that to the revised paper.
2. Figure 2 shows results presented in Cheng et al 2018, but we agree that we should add some more comments on details, even if they are provided in the previous paper. As for the dispersion relationship, please see our reply to your comment in part A.